# Assessing Multi-Hazard Vulnerability and Dynamic Coastal Flood Risk in the Mississippi Delta: The Global Delta Risk Index as a Social-Ecological Systems Approach

Carl C. Anderson [1,2,*](ID), Fabrice G. Renaud [3](ID), Michael Hagenlocher [2](ID) and John W. Day [4]

1   School of Interdisciplinary Studies, University of Glasgow, Dumfries, Scotland DG1 4UQ, UK
2   Institute for Environment and Human Security (UNU-EHS)—United Nations University, UN Campus, Platz der Vereinten Nationen 1, 53113 Bonn, Germany; hagenlocher@ehs.unu.edu
3   School of Interdisciplinary Studies, University of Glasgow, Dumfries, Scotland DG1 4ZL, UK; Fabrice.Renaud@glasgow.ac.uk
4   Department of Oceanography & Coastal Sciences, College of the Coast & Environment, Louisiana State University, Baton Rouge, LA 70803, USA; johnday@lsu.edu
*   Correspondence: c.anderson.4@research.gla.ac.uk

**Abstract:** The tight coupling of the social-ecological system (SES) of the Mississippi Delta calls for balanced natural hazard vulnerability and risk assessments. Most existing assessments have approached these components in isolation. To address this, we apply the Global Delta Risk Index (GDRI) in the Mississippi Delta at high-resolution census tract level. We assess SES spatial patterns of drought, hurricane-force wind, and coastal flood vulnerability and integrate hazard and exposure data for the assessment of coastal flood risk. Moreover, we compare current coastal flood risk to future risk in 2025 based on the modelled effects of flood depth, exposure, and changes in ecosystem area in the context of ongoing efforts under the 2017 Louisiana Coastal Master Plan. Results show that the Master Plan will lead to decreases in risk scores by 2025, but the tracts that are currently the most vulnerable benefit less from risk reduction efforts. Along with our index output, we discuss the need for further advancements in SES methodology and the potential for catastrophic hazard events beyond the model parameters, such as extreme rainfall events and very strong hurricanes. Assessing SES risk components can lead to more targeted policy recommendations, demonstrated by the need for Master Plan projects to consider their unequal spatial effects on vulnerability and risk reduction.

**Keywords:** vulnerability; disaster; census tract; GDRI; environmental hazard; storm surge; drought; hurricane; future risk

## 1. Introduction

Understanding the risk profiles of river deltas is important given their unique challenges, resulting from a high coupling of social-ecological systems (SES) with diverse populations and environmental characteristics [1–4]. Wetlands and intact coastal ecosystems not only support livelihoods but can also reduce the impacts of flooding and storm surge by acting as buffers [5–9]. Ecosystem services are themselves threatened by socionatural hazards, particularly those that are climate-sensitive [10]. A reinforcing feedback loop thus emerges of environmental degradation leading to diminished services and subsequently increased exposure [11]. The variety of hazards relevant to deltas can also have cascading impacts [12]. For example, drought may reduce the health of protective coastal vegetation, increasing coastal flood risk.

The risk of storm surge impacts in particular is increasing globally as a result of climate change, trends in coastal development, and associated sea level rise [13,14]. Over the past 200 years, storm surges have caused an average of 13,000 deaths per year [15], with some 250 million coastal residents currently exposed [16]. Meanwhile, global adaptation to climate change and associated increased hazard magnitude is limited by the

decrease in adaptive capacity of ecosystems resulting from climate change [17]. Healthy ecosystems that provide services operate within threatened temperature and precipitation boundaries [18]. Additionally, human disturbance of terrestrial and marine ecosystems contributes to increased risk [17,19]. In the Mississippi Delta, large navigation channels impede ecosystem restoration efforts, which must be integrated into existing interests in agriculture, urban development, fishing, and natural resource extraction [20]. Increases in hurricane magnitude can severely damage protective ecosystems, such as forested wetlands in the delta [21]. As wetlands are degraded, storm surge is able to travel farther inland [22]. While efforts towards dynamic modeling of surges is improving our ability to monitor the height, duration and distribution range of events and their impacts, limitations still exist [15,23] and there is a continued urgent need for risk assessment. Social and environmental factors must be considered, since areas most at risk from storm surge are also those that suffer from a continued reduction in natural resources combined with rapid urbanization [13].

The Mississippi Delta, like other global deltas, has a relatively large population that relies on a diverse range of ecosystem services stemming from a nexus of interacting environments [8,24]. The history of anthropological intervention with the Mississippi River and the delta that it feeds is emblematic of this diverse interaction between society and environment [6,7]. Increased hazard exposure has been observed as a result of environmental degradation [7] in an already highly exposed region [25]. Additionally, the Mississippi Delta has been identified as one of the most vulnerable regions in the U.S. [25–28] and was classified as a "delta in peril" by Syvitski et al. (2009) due to high levels of relative sea-level rise [29]. Pressures from climate change combined with unsustainable energy-intensive management practices in the delta will increase risk and therefore the importance of risk reduction efforts [30,31].

Natural hazard-related ecosystem assessments in the delta have mostly been limited to studies of land-loss or post-disaster damage (e.g., 19–21). Projected future shifts in exposure and risk profiles [2,28], partially a result of disproportionate impacts of climate change on deltaic systems [8,28,30,32], increase the urgency with which such assessments are needed in the Mississippi Delta. Moreover, the potential interactions and cascading impacts among hazards in deltaic SES calls for multi-hazard assessments [12], which allow for more tailored policy recommendations since risks are often hazard-dependent [33].

Devastating impacts from major hazard events in the delta in the 21st century along with the interconnected effects of climate change, land subsidence, and human-induced degradation led to the creation of Louisiana's Comprehensive Master Plan for a Sustainable Coast [34]. The "Master Plan" is revised in five-year increments to integrate scientific evidence with policy under the direction of The Coastal Protection and Restoration Authority (CPRA) [35]. The current version (2017) includes 124 projects at an approximate cost of 50 billion USD with three primary aims—build and maintain land, reduce flood risk and support ecosystems [34].

The Master Plan and its continued evolution is central for sustainable risk reduction efforts in the Mississippi Delta. Comprehensive scenario modelling efforts conducted in support of the Master Plan [36] allow for the combination and comparison of SES index-based risk assessment scores using the GDRI. Therefore, in addition to addressing the need for a coupled index-based risk assessment in the delta, we map spatially-explicit changes in risk scores by 2025 to determine what areas of the delta see the greatest benefits from ongoing risk reduction efforts. Providing equitable risk reduction benefits and reducing risk where it is most severe, while accounting for future changes, will be crucial for achieving the aims of the Master Plan.

To address the outlined gaps and ongoing challenges, we apply the GDRI in the Mississippi Delta at census tract level with three primary objectives: (1) address the need for a balanced SES assessment in the delta and provide a detailed risk profile as a tool for informing risk reduction, (2) determine the effects of the Master Plan on future social and

ecosystem risk and (3) advance spatial indicator-based SES risk assessment methodology through a critical reflection and discussion of results.

## 2. SES Risk Assessment: Concepts and Approaches

We define risk as the product of hazard characteristics combined with exposure and vulnerability [13]. Exposure denotes a spatial interaction of hazards with elements of value while the vulnerability of those elements modifies the severity of negative impacts. The three risk components can be demonstrated using Hurricane Katrina in 2005 as an example, which devastated the Mississippi Delta region [37]. The magnitude of Katrina and related coastal flooding represents the hazard component, the many residents as well as ecosystems which provide services in the delta area represent exposure, and the societal and ecological characteristics that modulated the severity of its impacts represent vulnerability. Assessing risk as the product of these components is a crucial step towards informing policy designed to avoid, mitigate, or transfer risk, as underscored by the Sendai Framework for Disaster Risk Reduction 2015–2030 [38]. One of the most common assessment approaches is the use of composite indicators or index-based approaches for the synthesis of multi-dimensional concepts like risk [39–41]. Indexes allow for the reduction of complexity through comparative rankings or scores, a useful characteristic for understanding and delivering messages in order to inform policy, foster debate or dialogue, and raise awareness [42].

Advancements have been made in vulnerability and risk research to recognize the interconnection of social and ecological systems through the concept of social-ecological systems (SES) [12,43–45]. In this research, we define SES as the spatial arrangement of interacting social and ecosystem characteristics in a spatially bound unit or regional area that modulate risk. The relation between environmental degradation and risk has been advanced within international conventions and frameworks such as the Hyogo Framework for Action in 2005, referring directly to "environmental vulnerabilities" [46]. In 2012 at the United Nations Conference for Sustainable Development or "Rio+20", the nexus of environment, disaster and sustainable development was once more emphasized as well as the role of healthy ecosystems [47]. Most recently, the Sendai Framework for Disaster Risk Reduction 2015–2030 (SFDRR) states that, "disasters significantly impede progress towards sustainable development" [38] and, in order to better understand disaster risk (Priority 1 of the SFDRR), it is important to "assess disaster risks, vulnerability [. . .] and their possible sequential effects [. . .] on ecosystems" [38]. By conducting risk assessments on this basis, not only risk reduction but also mitigation and adaptation strategies are broadened and sustainable ecosystem-based solutions to disaster risk reduction or climate change adaptation may be considered [47,48].

Despite the increasing recognition of their importance, balanced spatially explicit assessments of coupled SES vulnerability and risk are scarce [49,50]. Several attempts include work by Moss et al. (2001), who included ecosystem sensitivity but limited its assessment to only two proxy variables—percentage of land managed and fertilizer use [51]. Depietri et al. (2013) applied the MOVE (Methods for the improvement of vulnerability assessment in Europe) vulnerability assessment framework [52] to consider ecosystem services along with social vulnerability in the context of heat waves in the area of Cologne, Germany [53]. More recently, Calil et al. (2017) created the Comparative Coastal Risk Index and applied it along international coasts of Latin America and the Caribbean [54]. However, only ecosystem exposure was considered, but not vulnerability. Studies assessing vulnerability to climate change have taken a coupled SES perspective. O'Brien et al. (2004), for example, considered coupled human and environmental spheres in a study mapping vulnerability to climate change impacts in India [55], and Rani et al. (2015) also conducted an assessment within India but focused on coastal SES vulnerability [56]. The vulnerability assessment framework by Turner et al. (2003) has been operationalized in the context of such studies [32]. Luers et al. (2003) applied it to agriculture dimensions in Mexico [57], Westerhoff and Smit (2009) to climate change variability in Ghana [58], and Damm (2010)

used an adapted version to assess SES vulnerability to flooding in Germany [59]. Asare-Kyei et al. (2017) assessed SES multi-hazard risk in West Africa using an index-based approach [60].

Although there is a general lack of recognition of coupled SES in vulnerability and risk assessments, the gap in research is particularly critical for delta environments [48]. In their literature review of vulnerability assessments for three major global deltas, Sebesvari et al. (2016) found that, within SES indicator-based vulnerability assessments, a disproportionate number of social indicators are used compared to few ecosystem indicators [12]. In their study, Hagenlocher et al. (2018) introduce the Global Delta Risk Index (GDRI) concept and methodology based on a conceptual SES risk framework by Sebesvari et al. (2016) and applied it in the Mekong, Ganges-Brahmaputra-Meghna (GBM) and Amazon Deltas at the sub-delta scale [12,48]. More recently, a comparative study of the vulnerability dimension of the GDRI with the Social Vulnerability Index (SoVI®) in the Mississippi Delta was carried out at sub-delta scale [61]. However, the research centered on vulnerability index comparison and the implications of divergent theories, indicators and aggregation methodologies between the indexes on vulnerability scores. In this paper, we go beyond vulnerability to assess dynamic SES risk of storm surge and concentrate on the spatial distribution of vulnerability and risk scores in the Mississippi Delta. Additionally, we shift the focus of vulnerability and risk assessment in the delta towards a comprehensive set of ecological indicators to correspond with the range of social indicators used in past assessments [25–27,62]. This leads to a critical discussion of SES risk assessment and the challenges of assessing and confronting future risk in the delta.

## 3. Methods

### 3.1. Study Area

The Mississippi Delta is exposed to a range of natural hazards, some of the most relevant being droughts, flooding, hurricane winds and surge, storms, extreme precipitation events, and sea level rise [25,27,63]. Studies have also shown that the delta's highly anthropologically-altered ecosystems have become increasingly degraded, leading to diminished ecosystem services and higher levels of vulnerability and exposure in a changing climate [6,7,28,30]. Recent hurricanes in particular have devastated the region, notably Hurricane Katrina in 2005 resulting in over 1500 fatalities and causing the loss of 100 km² of wetlands in and around the Mississippi Delta [20]. Moreover, extreme precipitation events and increasing Mississippi River discharge threaten existing flood control systems [30,63].

For this study, the Mississippi Delta extent was delineated using the Coastal Zone Inland Boundary defined by the State of Louisiana Department of Natural Resources (http://www.dnr.louisiana.gov/index.cfm?md=pagebuilder&tmp=home&pid=928, accessed on 29 December 2020) and the assessment conducted at (sub-county) census tract resolution. We use census tracts because this is the highest spatial resolution data available and it allows us to assess spatial vulnerability and risk patterns within sub-delta regions, such as neighborhoods in the city of New Orleans. Using high-resolution spatially explicit units also has drawbacks since neither SES nor hazard characteristics are clearly spatially delineated, a point returned to in Section 5. Tracts are designed to contain the same average number of citizens and their varying sizes thus reflect population density. Census tracts fall within 16 parishes (synonymous with counties in other US states) of Louisiana (Text S1). A total of 457 tracts are assessed (Figure 1).

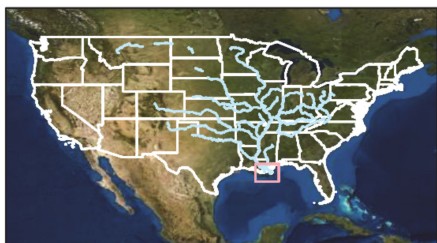

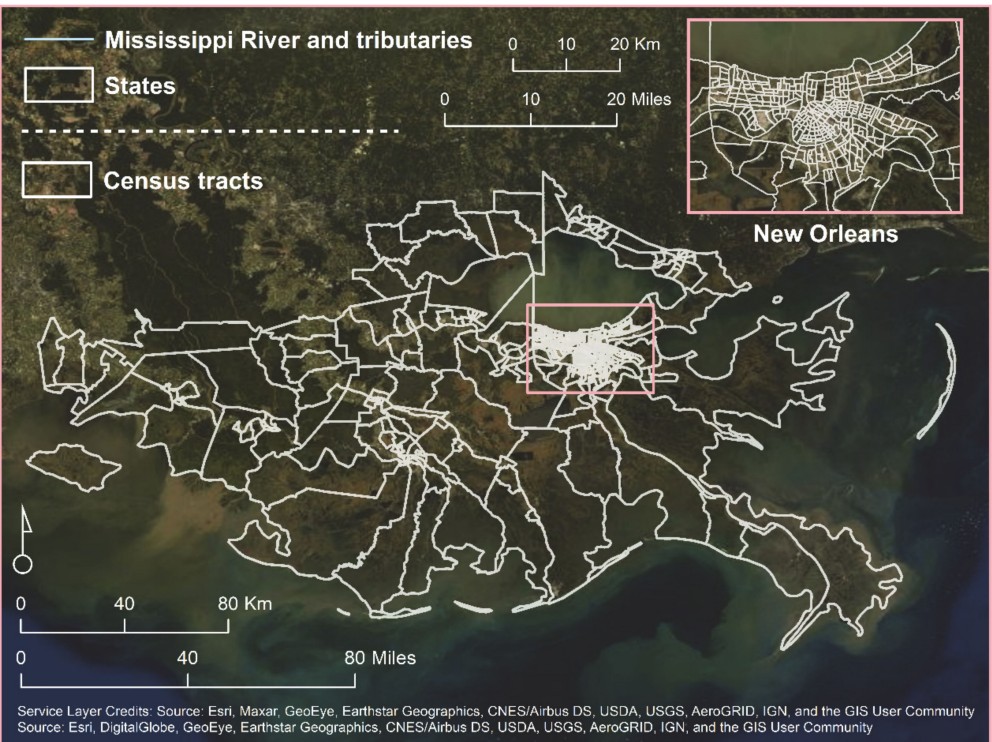

**Figure 1.** Study area map of census tracts (U.S. Census Bureau 2016) [64] in the Mississippi Delta included, based on the Louisiana State Coastal Zone Boundary (not shown).

### 3.2. GDRI Assessment of Multi-Hazard Vulnerability and Coastal Flood Risk

We apply the GDRI to assess SES multi-hazard vulnerability and coastal flood risk at census tract (sub-county) resolution. Because the GDRI is designed with a hierarchical modular methodology for calculating risk, it can be disaggregated into subcomponents along the divisions of social/ecosystem and hazard/exposure/vulnerability. Within vulnerability, the approach further differentiates along the conceptual lines of social/ecosystem susceptibility and lack of coping capacities [48]. This enables a thorough examination of the underlying drivers of risk scores. The GDRI also addresses the need for calculating hazard-specific assessment scores. Based on data availability and importance to the Mississippi Delta in terms of economic value as well as lives and livelihoods [25,27], we assess vulnerability to flooding from storm surge (i.e., coastal flooding), hurricane force wind, and drought. Next, we analyze coastal flood risk by combining vulnerability scores with the other two risk components of hazard (flood depth) and exposure (flood extent) scores. We do not assess risk to other hazards due to a lack of high resolution data to differentiate scores among census tracts.

Indicators for the vulnerability component of the assessment were selected from a library of deltaic indicators compiled in an extensive literature review by Sebesvari et al. (2016) on the basis of contextual relevance to the Mississippi Delta SES and data availability [12]. Thirty-two total indicators were used and grouped into four subcomponents—social susceptibility (n = 8), (social) coping capacity (n = 6), ecosystem susceptibility (n = 13), and ecosystem robustness (n = 5) (Table 1). These sub-divisions of vulnerability are most

closely based on work by Damm (2010) [59], later taken up in Sebesvari et al. (2016) [12]. Damm (2010) uses 'susceptibility' as a synonym for sensitivity, which is influenced by stressors on SES. "Capacities" are defined by the ability of a system to face hazard events and therefore also respond and recover. Ecosystem "robustness" was incorporated to correspond with social coping capacity, the former defined as the "capacity of the ecological system to absorb and resist disturbance while reorganizing and undergoing change" [59].

**Table 1.** Vulnerability component indicators used in the assessment arranged by subcomponent (social susceptibility, social coping capacity, ecosystem susceptibility, and ecosystem robustness). For hazard-specific scores, the indicators' relevance and cardinality ((+) for increasing vulnerability and (-) for decreasing vulnerability) for each hazard type was determined in a literature review by Sebesvari et al. (2016) [12]. Hazard types include hurricane force wind (H), drought (D), and coastal flood (CF). Proxies were used when their indicator data were more relevant to the Mississippi Delta context or otherwise not available.

| | Indicator | Hazard [1] | | | Proxy Taken | Time Period | Data Provider |
|---|---|---|---|---|---|---|---|
| | **Social Susceptibility** | **H** | **D** | **CF** | | | |
| 1 | % of the population with disabilities | + | + | + | N/A | 2014–2018 | American Community Survey (ACS) |
| 2 | % of illiterate population | + | + | + | % population over 25 without high school diploma | 2014–2018 | ACS |
| 3 | % of population below national poverty line | + | + | + | N/A | 2014–2018 | ACS |
| 4 | Dependency ratio | + | + | + | Age dependency ratio | 2014–2018 | ACS |
| 5 | GINI index (0–100) | + | + | + | N/A | 2014–2018 | ACS |
| 6 | Dependency on agriculture/forestry/fisheries for livelihood | + | + | + | % employed in farming, fishing, forestry and hunting, mining | 2014–2018 | ACS |
| 7 | % of population living in poorly-constructed houses | + | | + | % housing units mobile homes | 2014–2018 | ACS |
| 8 | % of households without an official land title / secure residential status | + | + | + | Occupied housing units—% renter-occupied | 2014–2018 | ACS |
| | **Social Coping Capacity** | **H** | **D** | **CF** | | | |
| 9 | % of households without access to information | + | + | + | Occupied housing units—no phone in household (cell or landline) | 2014–2018 | ACS |
| 10 | Access to shelter places | - | | - | N/A | 2016 | Federal Emergency Management Agency (FEMA) |
| 11 | Density of emergency services: hospitals, fire brigades, police stations[2] | - | - | - | N/A | 2017 | United States Geological Survey (USGS) |
| 12 | Density of transportation network[2] | - | - | - | N/A | 2014 | Environmental Protection Agency (EPA) |
| 13 | % of households without individual means of transportation: car or motorcycle | + | + | + | Occupied housing units no vehicles available | 2014–2018 | ACS |
| 14 | % of population without health insurance | + | + | + | N/A | 2014–2018 | ACS |

**Table 1.** *Cont.*

| | Indicator | Hazard [1] | | | Proxy Taken | Time Period | Data Provider |
|---|---|---|---|---|---|---|---|
| | **Ecosystem Susceptibility** | H | D | CF | | | |
| 15 | % of wetlands drained (wetland loss) | | + | + | N/A | 1932–2010 | Couvillion et al. (2011) [65] |
| 16 | Freshwater scarcity | | + | | Global fresh water resources | 2014 | Dickson et al. (2014) [66] |
| 17 | % of deforested area | + | + | + | N/A | 2000–2018 | Hansen et al. (2013) [67] |
| 18 | % of shoreline eroded | + | | | Average shoreline loss in meters | 1973–2001 | USGS |
| 19 | Wetland connectivity | | - | - | N/A | 2013 | U.S. Fish and Wildlife Service |
| 20 | River connectivity | | - | - | N/A | (1) 2011(2) 2016(3) 2016 | (1) EPA (2) U.S. Census Bureau (3) United States Army Corps of Engineers |
| 21 | Forest connectivity (RS) | | - | - | N/A | 2014 | Hansen et al. (2013) [67] |
| 22 | Water quality of freshwater bodies | | - | - | N/A | 2015 | EPA |
| 23 | Return flow ratio | | + | + | N/A | 2014 | WRI Aquaduct |
| 24 | Soil organic matter | | - | - | N/A | 2013 | SoilGrids |
| 25 | % of area covered by "problem soils" | | + | | N/A | 2012 | United States Department of Agriculture (USDA) |
| 26 | % of area covered by critical sites for conservation (danger of extinction) | + | + | + | Priority for conservation index | 2015 | Jenkins et al. (2015) [68] |
| 27 | Species richness adjusted by intactness | - | - | - | N/A | 2015 | Jenkins et al. (2013) [68] |
| | **Ecosystem Robustness** | H | D | CF | | | |
| 28 | % of forest area protected and designated for the conservation of biodiversity | - | - | - | % of protected area | 2014 | Jenkins et al. (2015) [68] |
| 29 | % of wetlands restored | - | - | - | N/A | 1985–2009 | Couvillion et al. (2011) [65] |
| 30 | % of forest area restored | - | - | - | N/A | 2000–2014 | Hansen et al. (2013) [67] |
| 31 | Ecosystem Functionality Index (EFI) | - | - | - | N/A | 2010 | Freudenberger et al. (2012) [69] |
| 32 | Mean Species Abundance (MSA) | - | - | - | N/A | 2010 | Global biodiversity model for policy support—GLOBIO |

[1] H = Hurricane (wind), D = Drought, F = Flooding (storm surge) [2] Indicators 11 and 12 were later removed to due multicollinearity.

Social vulnerability indicator data were readily available using the US Census. However, data for ecosystem indicators were calculated using GIS processing in ArcGIS (ESRI, Redlands, CA, USA; v. 10.7.1) and occasionally combining multiple data sets and developing new methodologies. There is also a discrepancy in the years for which data were

available. The year 2015 represents the approximate median period for data used in this research, which corresponds to the coastal flood data we later integrate into the model. Extensive documentation along with a step-by-step guide to ecosystem indicator data acquisition and creation is provided as supplementary material (Text S2).

Data for all social indicators are thus defined at census tract level, while the resolution of ecosystem indicators varies. Considering that census tracts also vary in size, the same data are applied to all census tracts within each larger data zone (e.g., 1 km raster grid cells). To account for uneven geometries (i.e., data grid extents not aligning with census tract boundaries) when relevant, zonal statistics are used in ArcMap to determine the mean data value for each census tract.

Following the GDRI workflow, data pre-processing for indicators included outlier detection (scatter plots and 5% trimmed mean) and treatment (winsorization) and multicollinearity detection (correlation matrices and variance inflation factor (VIF)). The min-max standardization method was applied [70] and indicator cardinality adjusted for relevant indicators by taking inverse values so that all higher values equate to higher vulnerability. We use min-max standardization due to many indicators with skewed distributions and varying ranges, as well as the difficulties associated with defining standardization thresholds [70]. The applied adjustment of cardinality enables interpretation of a lack of coping capacity and a lack of ecosystem robustness, with higher values corresponding to higher vulnerability. Five indicators (3, 4, 6, 7, 22) were treated using winsorization (Table S1). Two indicators (Density of emergency services [11] and Density of transportation network [12]) were excluded due to correlations of r > 0.90 ($p < 0.05$) with the indicator Access to shelter places (10).

The four modular GDRI vulnerability subcomponents were calculated from equally-weighted standardized indicators. We use equal-weighing based on the assumption of equal importance of index subcomponents. However, weights can easily be applied prior to aggregation based on evidence of differential indicator importance or to tailor the index for specific contexts, for example if a risk reduction intervention aims to primarily target ecosystem susceptibility, a corresponding stronger weight could be applied to that vulnerability subcomponent. Scores were combined to derive composite ecosystem vulnerability and social vulnerability scores that were, in turn, combined to derive SES vulnerability scores.

Going beyond vulnerability, only coastal flood risk was calculated due to a lack of variation in drought and hurricane force wind hazard datasets at census tract level. For coastal flooding, at the highest aggregative level are SES hazard, SES exposure, and SES vulnerability scores, the product of which yields final SES risk scores. SES hazard refers to the average flood magnitude (depth) within a tracts' ecosystem area and total area. We also derive social and ecosystem risk scores for flooding, framing index outputs around these components. Despite the delta being a SES, providing both social risk and ecosystem risk scores can inform targeted risk reduction efforts, which often focus on either ecological or social risk, rather than their interconnected combination. To calculate flood risk, only those indicators relevant to flooding are aggregated, while all indicators are used for multi-hazard vulnerability scores (Figure 2; Data S1).

All GDRI outputs are mapped using ArcMap 10.7.1 based on a quantile classification of relative scores with the classes of Low, Medium Low, Medium, Medium High, and High. We also use a No risk class for flood risk when there is no exposure (or corresponding hazard magnitude). Using quantiles, the same number of tracts are assigned to each class (n = 91) regardless of score distributions. Quantiles were chosen because of clustered final scores and to visually emphasize spatial patterns of relative risk in the delta, since the GDRI produces relative scores. It is important to note that e.g., Low risk is not an objective or absolute classification but must be interpreted as the lowest (in the bottom quantile) risk area in the delta according to the index. We return to this crucial point in the discussion by reflecting on the absolute risk of catastrophic flood events in the delta.

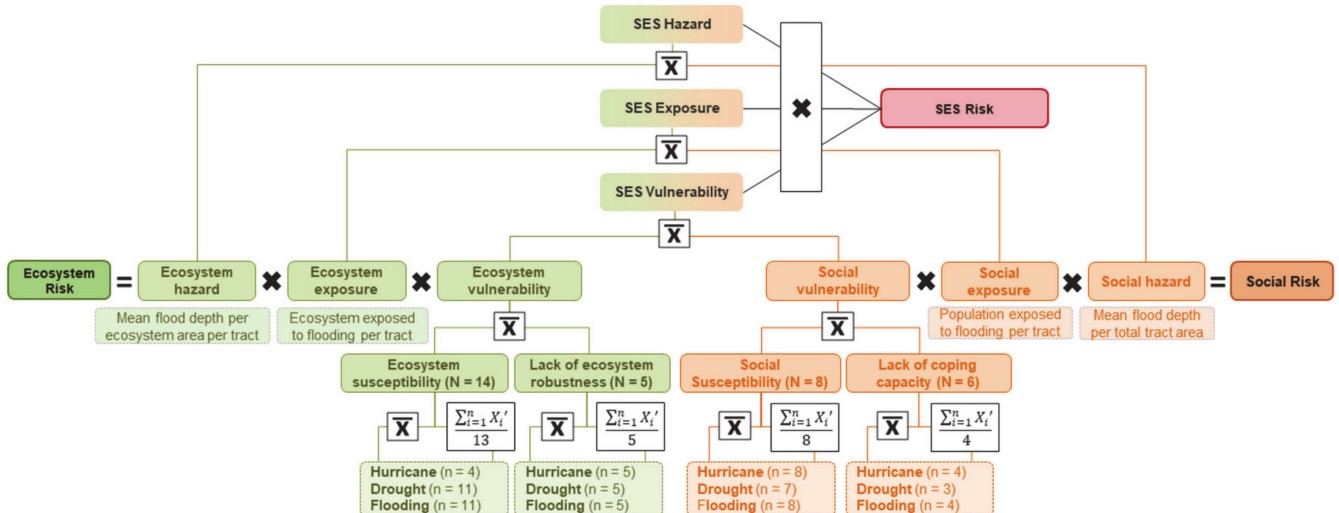

**Figure 2.** Aggregative modular structure of Global Delta Risk Index (GDRI) with aggregation occurring from the bottom up for each risk component—hazard, exposure, and vulnerability—generating intermediary scores for each subcomponent. Multi-hazard scores always use arithmetic means ($\overline{X}$) of all indicators within any given subcomponent. Hazard-specific scores use means except for the calculation of the first four vulnerability subcomponents, for which the varying counts of hazard-relevant standardized indicator values are summed and divided by the total number of indicators in that subcomponent. Social risk and ecosystem risk are calculated as the products of social (or ecosystem) hazard, exposure and vulnerability. Adapted from (Hagenlocher et al., 2018) [48].

Current and Future Coastal Flood Risk Calculation

To calculate current coastal flood risk (using data ranging from 2014–2018), we multiply flood vulnerability scores by flood exposure and hazard scores (Figure 2). We only assess coastal flood risk since exposure and hazard data for hurricane force wind and drought are excluded due to low spatial resolution. Variation in flood exposure among the tracts is a result of spatial overlap to any magnitude of flood, while 'flood hazard' varies according to the magnitude of flooding, defined here by flood depth. We assess both social risk to coastal flooding and ecosystem risk to coastal flooding as well as combining these scores into SES flood risk, taking the arithmetic mean score of each component per tract.

We use 500-year storm surge data from The Coastal Protection and Restoration Authority of Louisiana (CPRA) developed through the 2017 Coastal Master Plan modeling analysis [36] (Table 2). 500-year (0.2% probability of exceedance) data are used since the delta area has experienced several major events within the last decade and climate change will continue to usher in previously-considered "unprecedented" events [19,30]. Moreover, we assess relative risk in the delta and therefore the model is more useful for distinguishing differences ranging from low to high risk, rather than the majority of tracts being described as "0" risk due to no exposure under 10-year and 100-year flood models.

For social exposure, tabular population data per tract [71] were used given the greater accuracy and finer average spatial resolution than available spatial data. The number of people exposed to coastal flooding was determined by calculating the percentage of tract area exposed and taking the same percentage of total tract population. An even spatial population distribution is thus assumed, a simplistic but reliable method given that tract spatial delineation is itself based on population counts. To calculate ecosystem exposure, the percent vegetative land cover per tract (i.e., not classified as bare ground) exposed to coastal flooding was determined. Coastal vegetation data under the 'initial condition' (current) is also taken from CPRA Master Plan modelling [36,72,73]. This is a simplified proxy for ecosystem service provision (see NOAA's Coastal Flood Exposure Mapper (https://coast.noaa.gov/floodexposure/#/services (accessed on 21 November 2020)); 54), the limitations and implications of which are addressed in Section 5.

**Table 2.** Exposure and hazard component indicators used in the GDRI coastal flood risk assessment. Exposure and hazard data are combined with coastal flood vulnerability indicators (Table 1) to calculate coastal flood risk. Hazard data, also used to calculate social and ecosystem exposure, are integrated into the GDRI using 2015 data and modelled future scenario data to year 2025.

| | Indicator | Time Period | Data Provider |
|---|---|---|---|
| | **Social Exposure and Hazard** | | |
| 1 | % of the population exposed to coastal flooding | 2018 | American Community Survey (ACS) |
| 2 | 500-year event storm surge extent and depth | 2015; 2025 | The Coastal Protection and Restoration Authority of Louisiana (CPRA); Meselhe et al., 2017 |
| | **Ecosystem Exposure and Hazard** | | |
| 3 | % of the ecosystem area exposed to coastal flooding | 2015; 2025 | The Coastal Protection and Restoration Authority of Louisiana (CPRA); Meselhe et al., 2017 |
| 4 | 500-year event storm surge extent and depth | 2015; 2025 | The Coastal Protection and Restoration Authority of Louisiana (CPRA); Meselhe et al., 2017 |

Social hazard scores and ecosystem hazard scores follow a similar methodology. The former is calculated for each tract as the mean flood depth per total tract area (again assuming even population distribution), whereas ecosystem hazard scores are the mean flood depth per ecosystem area.

For future flood risk, we use CPRA Master Plan scenario modelling of flood extent and depth in the delta for a 500-year event under the 10-year 'medium' environmental scenario. This enables us to determine flood hazard and exposure reduction effects of the Master Plan and assess changes in risk per tract. We also use medium scenario data to 2025 of changes in vegetation (i.e., land cover) for calculating ecosystem components of risk in the GDRI. The modelled data have a baseline year of 2015 and the medium scenario (with values between the 'low' and 'high' future change scenarios) assumes moderate future changes in factors such as sea level rise, storm intensity and evapotranspiration [36]. We calculate social exposure based on the (relative) percentage of tract populations exposed, derived from changing flood hazard characteristics. Therefore, we do not integrate population change data since increases or decreases in absolute population values by 2025 will not affect the percentage-based index scores.

Since the original vulnerability data are maintained in this analysis, changes in risk are only a product of changes in the hazard and exposure components. We use the 10-year scenario (out to 2025) since changes in vulnerability are less likely to undergo dramatic alterations in spatial distribution and therefore this risk component can be included in the future comparison. We acknowledge the current limitation of not integrating future vulnerability data. However, since vulnerability is held constant, a clear picture of how flood risk will be affected by future changes in flood extent (used to derive exposure) and severity (hazard) is possible. Additionally, future scenario hazard data from 2015 are available only at 10, 25 and 50-year steps, while future vegetative land cover data are available at 10, 20, 30, 40, and 50-year steps. We therefore only assess risk out to 2025 (10-year step) since the next corresponding data describe year 2065 (50-year step). By 2065, there is too much uncertainty around exogenous model variables such as energy-intensive management practices in the delta [30,35] to confidently assess high resolution SES risk.

We first present changes in social and ecosystem delta exposure and hazard components out to 2025. We also show changes in the risk profile of the delta as a result of the Master Plan using social and ecosystem risk class changes per tract.

## 4. Results

### 4.1. Hazard-Specific Vulnerability

Hazard-specific vulnerability in the study area is visualized using the quantile symbology, yielding non-standardized scores ranging from 0.12–0.73 (Figure 3). Each map

shows relative vulnerability among the tracts for the respective hazard. Therefore, only the spatial distribution of low to high vulnerability classes can be compared across hazards or vulnerability types, but the actual numeric scores cannot. Variation in the maps is a result of several indicators that are not shared among the hazard types. For example, for social vulnerability the indicators Percent of population living in poorly-constructed houses (mobile homes) (7; Table 1) and Access to shelter places (10, Table 1) are not relevant to drought, while a number of ecosystem vulnerability indicators are not relevant to hurricane force winds (Table 1).

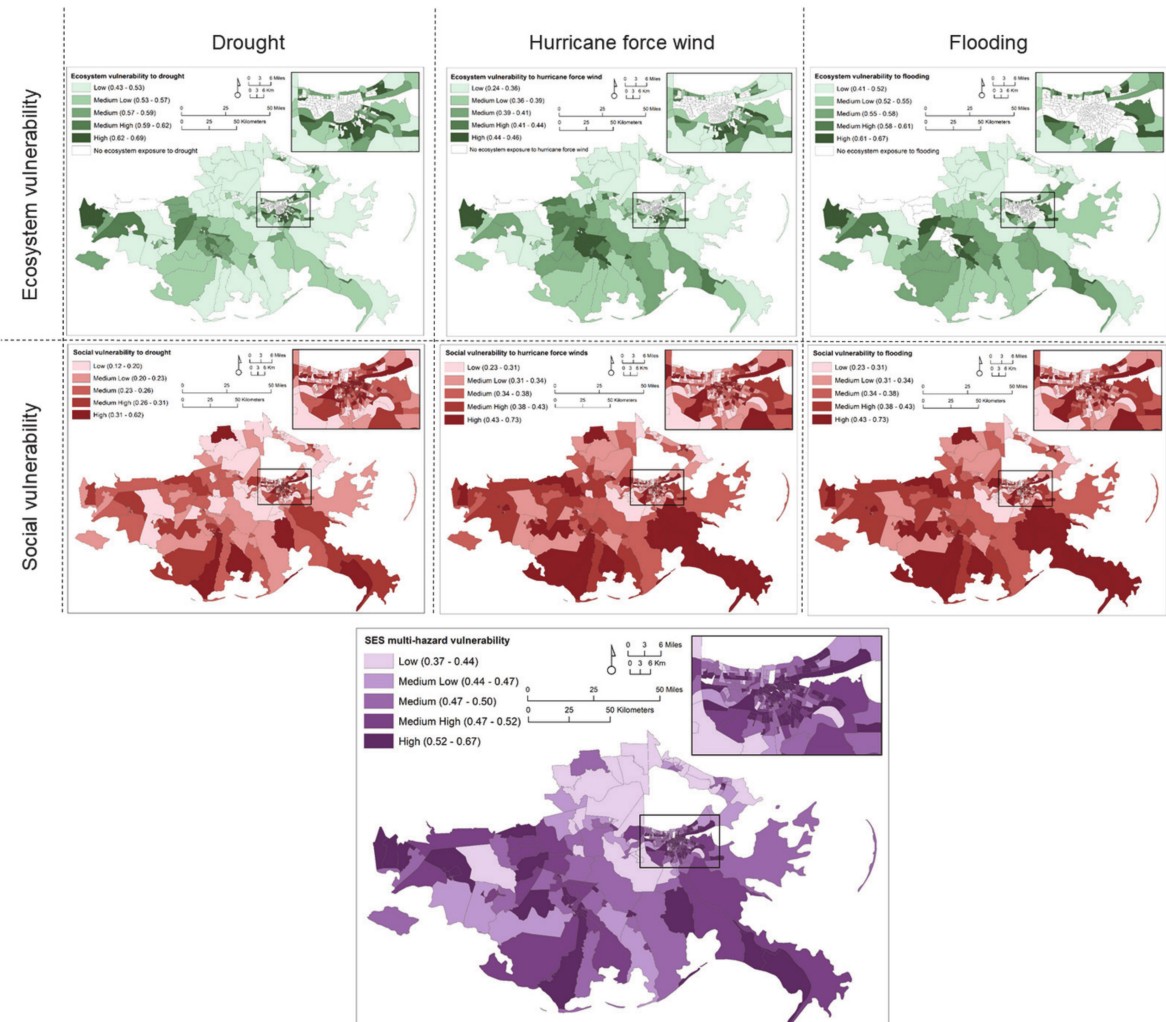

**Figure 3.** Social and ecosystem vulnerability for the hazards drought, hurricane force wind, and coastal flooding and Social-ecological System (SES) multi-hazard vulnerability. Ecosystem vulnerability is displayed only for those tracts in which there is some ecosystem exposure for the respective hazard. Because there is little ecosystem area among tracts in New Orleans, these are mostly shown in white (no ecosystem exposure).

Generally, both the peri-urban area around New Orleans and the area around Houma (central Western map area) shows higher ecosystem vulnerability. Disaggregating the ecosystem vulnerability scores into ecosystem susceptibility and ecosystem robustness shows high overlap between these subcomponents. Ecosystem susceptibility does contribute more to the area around New Orleans and down through the birdfoot delta (the Balize delta, extending farther down into the Gulf of Mexico), while ecosystem robustness accounts for more of the high vulnerability scores for the area around Houma.

In Figure 3, ecosystem vulnerability is only shown where there is some ecosystem exposure to the respective hazards. This creates a clear urban/rural divide defined by

ecosystem exposure, since the denser urban tracts are only composed of limited ecosystem area based on the underlying land cover data. However, ecosystem vulnerability in urban areas is on average higher than in rural areas given the effects of indicators such as Forest connectivity (21; Table 1), for example. The urban/rural discrepancy for ecosystem vulnerability highlights the importance of combining components all three components of risk for policy-oriented interpretation of results.

Social vulnerability does not show a clear urban/rural distinction, but there are hotspots of high vulnerability. Within New Orleans, there is high variability from neighborhood to neighborhood with generally higher vulnerability in the east and southeast of the city. Coastal tracts also consistently show high social vulnerability across hazard types. The tracts that compose the birdfoot delta are all in the highest vulnerability class for wind and flooding, with several Medium and Medium high tracts for vulnerability to drought. When disaggregating social vulnerability into its components of susceptibility and coping capacity, no prominent difference in spatial pattern emerges in the study area.

Combining social and ecosystem vulnerability for SES vulnerability, the larger, coastal tracts, specific clusters within New Orleans, and the northwest portion of the study area emerge as higher vulnerability. Contrarily, there is a cluster of low vulnerability tracts to the east of Lake Pontchartrain where there are few people (and thus larger tracts). Comparing urban versus rural tracts across the study area, only three of the total 33 indicators do not show a statistically different mean (at $p < 0.01$) for urban versus rural tracts, according to Mann-Whitney U tests (MW). These indicators all represent social disadvantage—Percent population below the poverty line (MW $p = 0.829$), Occupied housing units no phone in household (MW $p = 0.342$), and Percent of population without health insurance (MW $p = 0.474$). Therefore, clear differences emerge not only when comparing social and ecosystem vulnerability but also when comparing urban and rural areas of the delta.

### 4.2. Coastal Flood Risk

#### 4.2.1. Coastal Flood Hazard and Exposure

Our model shows that 230 of the 457 tracts in the delta are at risk of coastal flooding from storm surge, based on a modeled 500-year event (Figure 4).

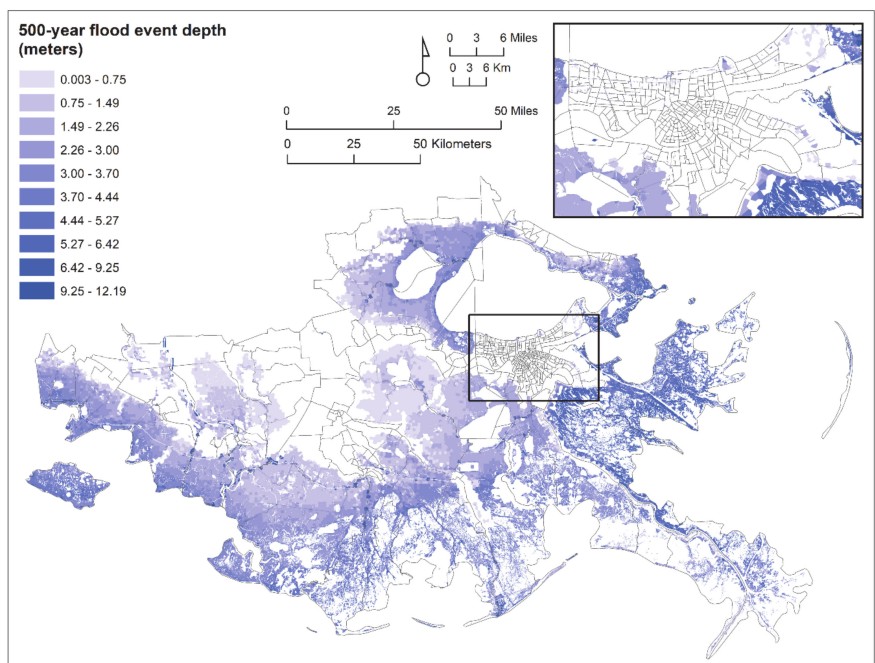

**Figure 4.** Extent and magnitude of coastal flooding for a 500-year flood event in the Mississippi Delta. Flood data is from Coastal Protection and Restoration Authority (CPRA) (2017) [34] and census tract data from (US Census, 2016) [64].

The mean flood depth per tract defines hazard strength in the model and ranges from 0 to 9.94 m (the maximum depth value is 12.19 m). The mean flood depth among all tracts in the study area is 0.28 m with a median of 0.04 m. Among only the 230 exposed tracts, the mean flood depth is 0.55 m with a 0.47 m median value. Combining the social and ecosystem hazard, vulnerability, and exposure scores yields scores for coastal flood risk in the delta (Figure 5).

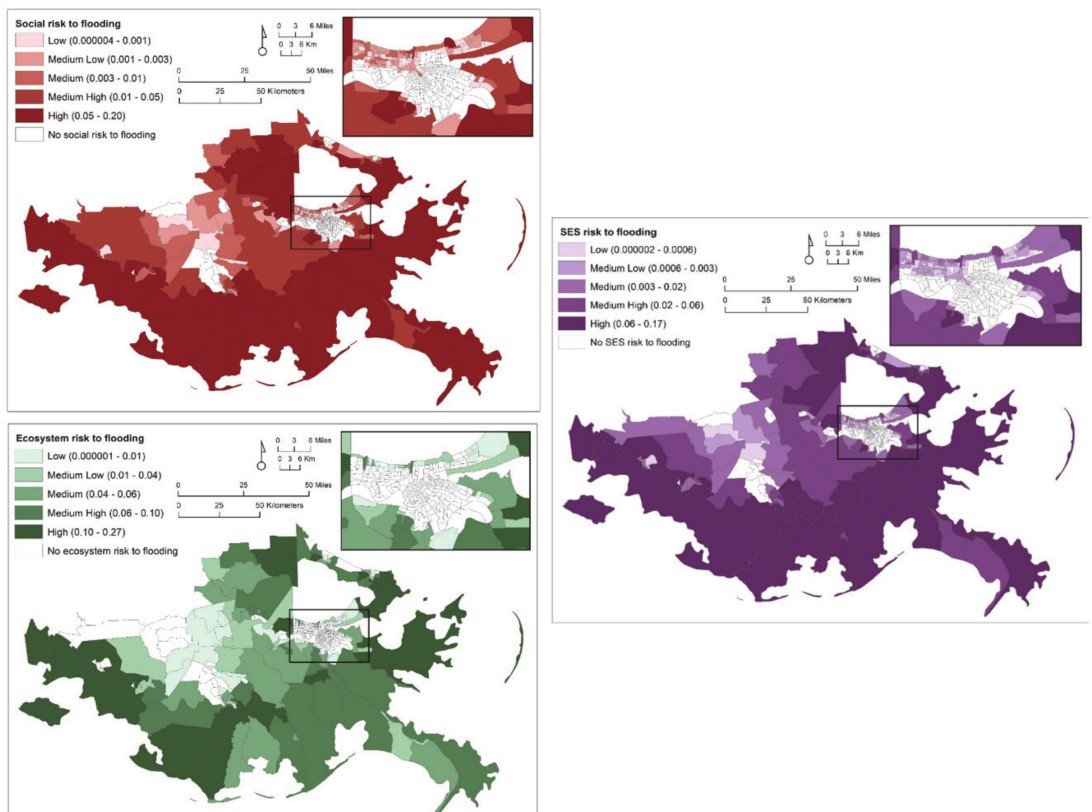

**Figure 5.** Social (top left) and ecosystem (bottom left) risk to coastal flooding (storm surge) in the delta. Averaging these scores yields SES flood risk scores (right). Flood data is from CPRA (2017) [34] and census tract data from (US Census, 2016) [64].

As expected, coastline tracts show the highest risk values, particularly for social risk. Flood exposure and magnitude is higher in these tracts and several vulnerability indicators also contribute to this pattern. Social vulnerability indicators that show the highest scores among coastline tracts include Dependency on agriculture/forestry/fisheries for livelihood, Percentage of population living in poorly-constructed houses [mobile homes], and Access to shelter places (6, 7, 10; Table 1). Ecosystem vulnerability indicators that show the highest scores among coastline tracts include Percentage of wetlands drained (wetland loss), Percentage of shoreline eroded, and Species richness adjusted by intactness (15, 18, 27; Table 1).

Because risk is calculated as the product of vulnerability, exposure, and hazard scores, 227 tracts (49.7%) have a No risk score for social risk (i.e., not exposed to flooding at all) and 317 tracts (69.4%) have No risk scores for ecosystem risk. The vast majority of null risk scores are for tracts within the city of New Orleans, where there is little to no ecosystem area defined (see Figures S1 and S2 for social and ecosystem exposure, respectively). The snapshot nature of the analysis means that shifting risk patterns and potentially inadequate flood defense systems against catastrophic events are not incorporated in the model. Of course, the current absolute risk is far from null. These limitations are taken up in the discussion.

4.2.2. The 2017 Louisiana Coastal Master Plan and Risk Reduction in Year 2025

Using the GDRI, we compare the current risk model with the future 2025 risk model that incorporates changes in coastal flood exposure and magnitude (hazard) and vegetation based on 2017 Master Plan projects. On average across the study area, all risk components and input data indicate decreases in risk (Table 3).

**Table 3.** Changes among Mississippi Delta census tracts comparing current ecosystem area data and current coastal flood area and depth data (from baseline 2015 data) with corresponding data for 10-year medium environmental scenarios. Average changes (absolute and relative) across the delta as well as the tract values for greatest absolute decrease and increase in risk components are shown.

| | Risk Component | Average Abs. Change | Greatest Abs. Decrease | Greatest Abs. Increase | Average % Change [1] |
|---|---|---|---|---|---|
| **Area exposed to flooding (km$^2$)** | N/A; input | −1.30 | −55.67 | 56.02 | −12.09 |
| **Population exposed to flooding[2] (count)** | Social exposure | −153.72 | −7809.00 | 531.30 | −11.84 |
| **Average flood depth (m)** | Social hazard | −0.04 | −5.52 | 8.11 | −10.76 |
| **Ecosystem area (km$^2$)** | N/A; input | −1.03 | −28.17 | 7.36 | −8.11 |
| **Ecosystem area exposed (km$^2$)** | Ecosystem exposure | −1.03 | −32.06 | 9.35 | −8.30 |
| **Ecosystem area flood depth (m)** | Ecosystem hazard | −0.07 | −4.28 | 7.04 | −5.74 |

[1] Average % change column displays values first treated for outliers. All values of >100% increase or <−100% decrease are not included in the average. [2] Calculated using proportion of area exposed by tract; does not account for spatial distribution of population within tracts.

On average, the delta tracts will lose about 1 km$^2$ of ecosystem area by 2025 despite the efforts of the Master Plan. Therefore, there will be less ecosystem area (8.11% decrease of ecosystem area on average), and it will be slightly less exposed to flooding (8.30% decrease in exposed ecosystem). Average flood depths decrease only very slightly, with an average drop of 0.04 m (4 cm) per total tract area and 0.07 m (7 cm) drop on average per tract ecosystem area. These figures equate to generally decreasing GDRI risk component scores in 2025, but with unequally distributed effects (Figure 6). The rather modest reductions in risk reflect the counteracting effects of climate change in the coastal zone (Day and Erdman, 2018), modelled under the CPRA medium scenario [36,73].

Risk does not change in most New Orleans urban tracts with the exception of decreases in social risk along the border of Lake Pontchartrain in the northwestern portion of the city. A total of 18 tracts move to No risk for social risk since they are no longer exposed, while 16 tracts increase in social risk. These include several tracts with scores that increase more than 100% near Houma and others in the area between Lake Pontchartrain and Lake Borgne, where flooding is expected to increase in extent and severity.

Generally, the most vulnerable rural tracts—coastal tracts, tracts in the southwestern peri-urban New Orleans area, and tracts around Houma (see Figure 3 for vulnerability maps)—do not see gains in risk reduction by 2025 under the conditions of the 2017 Master Plan. Ecosystem risk, in particular, increases in these tracts that are currently the most vulnerable. Conversely, the greatest gains in ecosystem risk reduction occur where vulnerability is low, among scarcely populated tracts to the northwest of New Orleans. Upcoming projects and plans in the delta should aim to counteract these potentially increased disparities in risk resulting from social and environmental changes combined with the 2017 Master Plan response.

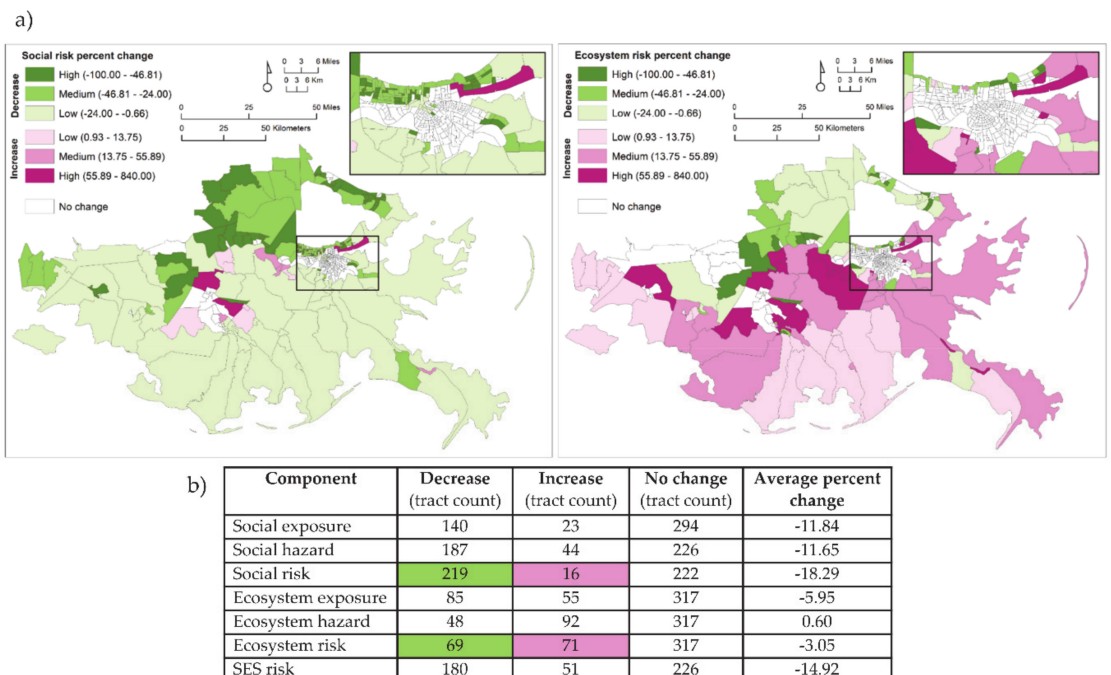

**Figure 6.** Percentage change in future social (*left*) and ecosystem (*right*) risk scores per tract in 2025. Darker shades of green indicate greater decreases in risk and darker shades of pink indicate greater increases in risk (**a**). The number of tracts in the delta that decrease, increase, or do not change in risk and risk component scores, as well as corresponding average percent change are also shown (**b**). The mapped elements in (**a**) are colored in the table (**b**) according to decreases in risk (green) and increases in risk (pink).

## 5. Discussion

The GDRI contributes to a shift in vulnerability and risk concepts towards more of a focus on coupled SES rather than only social characteristics of populations. The mutual dependency of human and environment is exemplified in deltaic systems [1–4,8,12,74,75] and supports the consideration of a coupled SES when assessing vulnerability and risk [28,48]. Mississippi Delta residents have been historically reliant on ecosystem services including water quality and supply as well as hurricane and flood protection [3,24], all stemming from the most biologically productive ecosystem in the U.S. [7]. However, a recognition of the delta as a coupled SES should serve to add complexity to our understanding of risk in the delta rather than obscure the importance of social vulnerability. Indicators from the GDRI exemplify the need to not detract from the social realities of the delta, demonstrated by on average 30% of over 25-year-olds without a high school diploma and 20% of the population living in poverty (Indicators 2 and 3; Table 1) [71]. Past events like Hurricane Katrina that have disproportionately negatively affected marginalized groups such as the poor, elderly, renters and blacks [76] further demonstrate the urgent parallel need for socially-oriented policy.

For ecosystem vulnerability, wetland and forest loss were found to be primary drivers (ecosystem robustness subcomponents). This loss is so great that 25% of Mississippi Delta swamps and marshes have become open water over the past century [7]. Marshes, mangroves and swamps can provide protection from storm surge and complement existing structural protection. Wetlands are crucial for trapping sediments and, in some circumstances, allowing the land to keep up with increasing sea level rise, while forested wetlands can also reduce wind penetration [6,77]. The potential of healthy wetlands as nature-based solutions to reduce the impacts of coastal flooding hazards is well-known [5,7,8,78] and its efficacy as a policy option reflected in the GDRI results. Mangrove restoration, for example, has been a common ecosystem-based response to reduce coastal flood risk by attenuating wave height and protecting coastal livelihoods [79]. Such measures are particularly valued

due to co-benefits such as increased food security and climate change mitigation through carbon storage [13,80].

Past and ongoing emphasis on risk reduction measures has centered on structural protection, both globally [81] and in the Mississippi Delta in the form of increasingly larger levee systems [3]. There is also a need for "soft" or non-structural options that are generally able to be implemented more quickly and at lower cost [19,82] along with a shift in emphasis towards nature-based solutions and environmental protection, which will help attenuate the reinforcing feedback loop of environmental degradation and risk from coastal hazards. Efforts have been appropriately augmented to restore particularly wetland habitat in the wake of recent major hurricanes in the Mississippi Delta [20]. The 2017 Master Plan allocates approximately 6.2 billion USD to non-structural measures over the next 50 years [19]. A number of nature-based solution approaches such as marsh creation and diversions [3] are expected to decrease the rate of land loss and make gains towards decreasing both social and ecosystem risk. However, a continued lack of sustainable management may undermine these efforts [30]. We show that environmental changes result in some areas with increases in risk by 2025 despite projects carried out under the Master Plan. Our results are in line with those of Fischbach et al. (2019) who find that the greatest gains in economic risk reduction occur to the southwest of New Orleans and north of Lake Pontchartrain from Master Plan projects [83]. However, our results do not highlight the area along the west bank of the Mississippi River with increased risk. Crucially, we find that the benefits of risk reduction are not equitable and, in some cases, may even increase current spatial disparities between low and high risk delta regions. Increased attention should be given to these potential disparities in upcoming delta projects and plans. In particular, refocusing risk reduction efforts on the areas with the highest current SES vulnerability levels is warranted. While flood magnitude and exposure reductions are important, underlying social and ecosystem vulnerabilities, particularly under conditions of increased social and ecosystem pressures, must be addressed for successful risk reduction. This finding also underscores the importance of further research to integrate future vulnerability scenario data for a more complete analysis [84,85].

Cascading hazard events are increasingly recognized as important considerations for successful risk reduction (e.g., [17,38]). For example, storm surge can degrade coastal ecosystems, thereby changing flood and drought patterns [86]. Assessing multi-hazard vulnerability should allow decision-makers to more realistically plan and implement measures to address this potentiality [18]. Of course, multi-hazard risk assessments should be strived for when data are available. Decision-makers must be made aware of the spatial distribution of risk for other relevant interacting hazards. However, more research is needed to better understand decision-making and public risk communication in the context of multiple hazards, continuing to move beyond information-deficit models [87]. For this, public perceptions of multi-hazard risk must also be understood and crowdsourced knowledge (e.g., from citizen science campaigns) integrated into the information portfolios of decision-makers. Such initiatives have proven particularly useful for ecosystem monitoring [88], such as wetland degradation and loss in the Mississippi Delta [89], which could help overcome the current limitations of ecosystem indicator data availability. Results of these efforts could feed into the public participatory opportunities provided by the upcoming 2023 Louisiana Coastal Master Plan and support strained local governmental capacities in the delta [90].

Operationalizing the theoretical advancements of a SES perspective is not without inherent limitations [32,44,91] and determining the degree to which human and ecological systems are coupled and therefore synergistically vulnerable can be complex and variable [31,92,93]. The simplistic calculation of ecosystem exposure applied in the GDRI represents the difficulty of defining what and where an ecosystem is. GDRI methodology and corresponding results in this assessment imply that urban areas are more vulnerable (at least on one dimension) than rural areas due to a dearth of surrounding ecosystem services. These assumptions may be more justified by explicitly accounting for access to-

and benefits from spatially contextual ecosystem services [50]. Instead, the current model configuration implies that the tracts relatively most affected by hazards have the highest vulnerability, rather than tracts with the (absolute) most to lose (for example, in terms of biodiversity). Moreover, different types of wetlands provide different ecosystem services. For example, cypress swamps perform best at reducing storm surge and waves [3]. Such detailed analysis should be taken up in future models, which will be further aided as data availability for high-resolution ecosystem indicators improves. Starting from an ecosystem service perspective and supplementing GIS with qualitative methods to determine the flow of benefits from ecosystems to delta residents, including who benefits, where, and how much, should be taken up in future research. The GDRI is designed to allow for integration of expert weighting, which would help to address SES-related limitations in future research, as well as indicator compensability (i.e., using arithmetical means, as one indicator increases in value other indicators are proportionally pulled upwards with it; 70). Such qualitative data provided by local stakeholders will need to be more heavily relied on to adequately capture the dimensions of SES risk and apply the GDRI in data scarce regions (e.g., 48). The indicators used in this study build on the useful indicator library provided by Sebesvari et al. (2016) [12] by applying the GDRI in the context of a developed (global north) delta.

Prior research demonstrated that the effects of index construction methodology can have even more influence on output scores than the theoretical starting point of assessing social, ecological, or balanced SES vulnerability in the delta [61]. In this work, convergent validity between the GDRI and SoVI® were weak, reducing confidence in output scores. Comparing the social vulnerability scores in the GDRI with those of the 2017 Coastal Master Plan [94] by using five classes from low to high vulnerability, we find good spatial overlap with 78.3% of tracts changing no class or one class (e.g., low to medium low; Table S2). Currently, without a similar study of high resolution ecological vulnerability in the delta it is difficult to determine the external validity of this assessment component, underlining calls for further research. However, it is certain that applying vulnerability scores to spatially explicit units (census tracts) within a delineated study area is a simplified model of reality, since interactions that occur across units and beyond study area boundaries can be influential [92,95]. While this has been addressed in SES risk frameworks, it has yet to be applied in a risk assessment [12,49]. The static, spatially explicit method does not account, for example, for the possibility of increased vulnerability in inland areas due to environmental degradation in rural coastal areas; e.g., the loss of livelihoods through supply chains. For this, methods from systems modeling, ideally based on qualitative stakeholder input, could be incorporated. Exploring the use of differential spatial buffers that follow human and environmental (rather than political) boundaries such as catchments or transportation infrastructure may be worthwhile. However, risk assessments must consider the inevitable trade-off of approximating reality by increasing complexity versus simplicity by reductionism [91], the latter of which is evidenced by the simple hierarchical structure of the GDRI.

We recognize that there are other important hazards in the delta, in particular the hazard data do not incorporate riverine flooding and extreme precipitation events. Likewise, vulnerability indicators are not all of equal importance and this varies spatially within the delta. The interaction between hazard and vulnerability data is likely non-linear and there exist certain thresholds not accounted for in the assessment. For example, at the upper end of the hazard data (i.e., storm surge magnitude), a truly catastrophic event in which flood defense fails may more equally affect those living either above or below the poverty line or those with or without a vehicle than a more moderate event. Likewise, for ecosystem vulnerability it is likely that, for example, wetland degradation is more important than soil organic matter. Again, expert-weighting would improve the model and make progress towards addressing these issues. Data availability would also improve the reliability of the model, since data are derived from a range of years, with a median year of about 2015. Integrating the full range of hazards with high resolution data and determining how

vulnerability indicators behave in relation to each other, across fixed time periods, would provide useful insight in future research into how vulnerability is changing in the delta.

The boundaries of our model parameters also reflect the trade-off of reducing complexity. In particular, the very real potential for catastrophic flood events is not captured in the 500-year flood data or the Master Plan modelling [36,63]. The potential for more frequent category 4 and 5 hurricanes that intensify faster, cover larger areas, move more slowly, and release more rainfall is increasing, and intense rainfall events are becoming more common [19]. For example, slow moving Hurricane Harvey dropped about 1.5 m of rain in three days over the city of Houston, also in the Gulf of Mexico. Hurricane Laura in 2020, which made landfall in southwest Louisiana, had devastating 240 km/h winds 65 km inland. If such a storm was centered on New Orleans it would likely have caused catastrophic flooding in the entire metropolitan area. This is the future for coastal Louisiana and such storms will almost certainly re-occur in this century. Therefore, the low risk scores produced by the model for urban tracts in New Orleans do not reflect the very real potential for future catastrophe. As these events continue to occur, hazard scenario data will more closely represent the future of the delta and allow for more accurate assessments.

Other interacting factors, including the combination of resource scarcity, an uncertain energy future, shifting populations and the ongoing Covid-19 pandemic will exacerbate risk in the Mississippi Delta and beyond. Reducing the risk of disasters, which is inhibited by global changes, is currently an intensively fossil fuel dependent activity. For example, Weigman et al. (2017) reported that the interaction of sea level rise and increasing energy costs on the cost of building marshes with dredged sediments led to dramatic increases in the cost of marsh creation [31], one crucial aspect of Louisiana's Coastal Master Plan [34]. Reducing risk proactively requires continued major investment, along with the need to properly plan for and respond to disasters when they occur in the delta.

## 6. Conclusions

By assessing risk from a SES perspective, the GDRI makes progress towards capturing the coupled nature of the Mississippi Delta. The efficacy of historically divergent yet equally appropriate efforts towards reducing social vulnerability and exposure and increasing ecological resilience can be leveraged when combined. This research builds on previous work comparing the SoVI® with the social vulnerability component of the GDRI by focusing on the spatial distribution of SES vulnerability, along with current and future storm surge risk in the delta. Effective risk reduction policies should consider interlinked elements within the complex deltaic system as well as the relative contributions of risk components (vulnerability, exposure and hazard) to overall risk.

Differences in spatial patterns of social and ecosystem risk reduction due to projects carried out under the 2017 Louisiana Coastal Master Plan reflect this need. Future planning and allocation of resources for risk reduction should consider both the effects on individual as well as combined risk characteristics. This will become increasingly necessary due to rates of social and climate change and the looming potential for catastrophic coastal flooding events in the delta. The hierarchical and modular construction of the GDRI, which allows for aggregation and decomposition, will prove useful in this regard.

Although the GDRI and its underlying SES framework signify progress, they do not represent a universal theory of risk but rather an important contribution to historic and current efforts at risk assessment and reduction in the Mississippi Delta. Moreover, better representing SES complexity in combination with the multi-dimensional and dynamic concept of risk calls for further research and method development. Nevertheless, the GDRI assessment presented provides empirical evidence through spatially explicit and modular risk scores, the proper interpretation of which can inform measures designed to reduce risk.

**Supplementary Materials:** The following are available online at https://www.mdpi.com/2073-4441/13/4/577/s1, Text S1. List of Louisiana parishes within the Mississippi Delta study, Text S2. Ecosystem vulnerability indicators—data acquisition and creation, Table S1. Outlier tracts per indicator treated using winsorization, Table S2. Social vulnerability sensitivity analysis, Data S1.

GDRI in the Mississippi Delta indicator data and aggregation, Figure S1. Social exposure to coastal flood in the Mississippi Delta, Figure S2. Ecosystem exposure to coastal flood in the Mississippi Delta.

**Author Contributions:** Conceptualization, C.C.A., F.G.R. and M.H.; methodology, C.C.A., M.H., F.G.R. and J.W.D.; data curation, C.C.A.; writing—original draft preparation, C.C.A.; writing—review and editing, C.C.A., F.G.R., M.H. and J.W.D.; visualization, C.C.A.; supervision, F.G.R. and M.H. All authors have read and agreed to the published version of the manuscript.

**Funding:** The research was part of the international Belmont Forum project BF-DELTAS "Catalyzing action toward sustainability of deltaic systems with an integrated modeling framework for risk assessment." UNU-EHS was funded in part by the German Research Foundation (DFG) (Grant no.RE 3554/1-1). The contribution of Carl C. Anderson was supported by a University of Glasgow College of Social Sciences PhD Scholarship.

**Institutional Review Board Statement:** Not applicable.

**Informed Consent Statement:** Not applicable.

**Data Availability Statement:** Data is contained within supplementary material.

**Acknowledgments:** We would like to thank Zita Sebesvari for her supervision, expert advice regarding ecological data acquisition and integration, and general support of this research. We would also like to thank the anonymous reviewers for their critical feedback that allowed us to improve the work.

**Conflicts of Interest:** The authors declare no conflict of interest. The funders had no role in the design of the study; in the collection, analyses, or interpretation of data; in the writing of the manuscript, or in the decision to publish the results.

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
