# Peer review of "Assessing Multi-Hazard Vulnerability and Dynamic Coastal Flood Risk in the Mississippi Delta: The Global Delta Risk Index as a Social-Ecological Systems Approach"

_water, doi:10.3390/w13040577_

Round 1

Reviewer 2 Report

A BRIEF SUMMARY

The paper titled “Assessing multi-hazard vulnerability and dynamic coastal flood risk in the Mississippi Delta: the Global Delta Risk Index as a social-ecological systems approach” presents a good topic for readers of this Journal. The topic represents a line of research as interesting as studied. The paper is well structured.

The results are carefully described and analysed in the paper. Hovewer, some question remain after reading the paper. Below is the list of some questions that need to be addressed.

  1. I have read this paper: “Anderson, C.C., Hagenlocher, M., Renaud, F.G., Sebesvari, Z., Cutter, S.L. and Emrich, C.T., 2019. Comparing index-based vulnerability assessments in the Mississippi Delta: Implications of contrasting theories, indicators, and aggregation methodologies. International Journal of Disaster Risk Reduction, 101128”. I believe it would be appropriate to specify the novelties of this manuscript with respect to your previous study. These novelties must also be reported in the conclusions. In my opinion, solved this question, the paper will be ready for publication; alternatively I will reject the manuscript, because too much similar with your previous work and without a substantial improvement.
  2. The authors have “storm surge” in keywords. I suggest two works, for your introduction, about this topic.
  3. Finally, I hope to read in the revised version suggestions for future developments of this interesting work.

SPECIFIC COMMENTS

I suggest the following studies on “storm surge” and “risk analysis”.

  • Qu, K., Yao, W., Tang, H.S. et al. Extreme storm surges and waves and vulnerability of coastal bridges in New York City metropolitan region: an assessment based on Hurricane Sandy. Nat Hazards (2021). https://doi.org/10.1007/s11069-020-04420-y
  • Apollonio, C.; Bruno, M.F.; Iemmolo, G.; Molfetta, M.G.; Pellicani, R. Flood Risk Evaluation in Ungauged Coastal Areas: The Case Study of Ippocampo (Southern Italy). Water 2020, 12, 1466. https://doi.org/10.3390/w12051466

Reviewer 3 Report

Notwithstanding the geographic focus of the work, it would be interesting to see the discussion expanded to reflect on other delta systems.

Author Response

Thanks very much for your review. Due to the length of the discussion, we rather add a sentence to refer the reader to work by Sebesvari et al. (2016), which serves as a useful starting point for the application of the GDRI in other deltas.

lines 653-655: The indicators used in this study build upon the useful indicator library provided by Sebesvari et al. (2016) [12] by applying the GDRI in the context of a developed (global north) delta.

Reviewer 4 Report

The time horizon for which changes in risk scores are evaluated (2025) is very close to the present moment (2021). A wider time horizon would have been more appropriate (2035 or for example), performing an interpolation between the data available for future scenario hazard data or for future vegetative land cover data.

There are also some minor corrections to be made:

Rows 349-351   “500-year (0.2% probability) data are used ……”

In fact, 500 year return period means 0.2% probability of exceedance (not probability of occurrence).

Row 638

“…..the tracts relatively most effected by hazards have the highest……”

It should be “affected”

Row 683

“……flood defense fails may more equally effect those living either…..”

It should be:

“……flood defence fails may more equally affect those living either…..”

Author Response

Thanks very much for your review. We agree that the interpolation of data sets into the future would likely be a good approach to the issue of mismatched scenario data. We appreciated that recommendation and will consider it for future work.

line 639 changed “effected” to “affected”

line 506 changed from “defense” to “defence”

line 684 changed from “defense” to “defence” and “effect” to “affect”

Round 2

Reviewer 2 Report

The paper has been improved following reviewer comments. In my opinion, it is ready for publication. Congratulations.

Author Response

Thanks very much for your review.